# Unravelling raked linear dunes to explain the coexistence of bedforms in complex dunefields

Ping Lü[1,2], Clément Narteau[3], Zhibao Dong[1,2], Olivier Rozier[3] & Sylvain Courrech du Pont[4]

Raked linear dunes keep a constant orientation for considerable distances with a marked asymmetry between a periodic pattern of semi-crescentic structures on one side and a continuous slope on the other. Here we show that this shape is associated with a steady-state dune type arising from the coexistence of two dune growth mechanisms. Primary ridges elongate in the direction of the resultant sand flux. Semi-crescentic structures result from the development of superimposed dunes growing perpendicularly to the maximum gross bed-form-normal transport. In the particular case of raked linear dunes, these two mechanisms produces primary and secondary ridges with similar height but with different orientations, which are oblique to each other. The raked pattern develops preferentially on the leeward side of the primary ridges according to the direction of propagation of the superimposed bed-forms. As shown by numerical modelling, raked linear dunes occur where both these oblique orientations and dynamics are met.

[1] Department of Geography, Shaanxi Normal University, 620 Chang'an West Avenue, Xi'an, Shaanxsi 710119, China. [2] Key Laboratory of Desert and Desertification, Northwest Institute of Eco-Environment and Resources, Chinese Academy of Sciences, 320 West Donggang Road, Lanzhou, Gansu 730000, China. [3] Institut de Physique du Globe de Paris, Sorbonne Paris Cité, Université Paris Diderot, UMR 7154 CNRS, 1 rue Jussieu, 75238 Paris Cedex 05, France. [4] Laboratoire Matière et Systèmes Complexes, Sorbonne Paris Cité, Université Paris Diderot, UMR 7057 CNRS, Bâtiment Condorcet, 10 rue Alice Domon et Léonie Duquet, 75205 Paris Cedex 13, France. Correspondence and requests for materials should be addressed to P.L. (email: lvping@lzb.ac.cn) or to C.N. (email: narteau@ipgp.fr).

Raked linear dunes display asymmetric morphologies with long primary ridges breaking down only on one side into periodic secondary ridges with a perpendicular alignment to the main crest and a semi-crescentic shape. This unique pattern provides an intriguing case study to begin to examine how primary and secondary bedforms with similar heights but different trends can coexist in the same wind regime.

A major question in arid zone geomorphology is to determine to what level active dunefields are in dynamic equilibrium with the current wind regime or the result of longer-term processes like climate change[1]. This is a challenging task given the broad spectrum of observed dune shapes, which are primarily controlled by sand availability and wind directional variability[2,3]. If crescentic barchans (low sand availability, unidirectional wind) and star dunes (high sand availability, multidirectional winds) occur under two extreme and opposite sets of conditions, linear dunes are often sub-divided into different categories according to their shape[4,5] (simple, compound and complex) or orientation with respect to the resultant sand flux direction[6,7] (transverse, oblique and longitudinal). Unfortunately, all these classifications fail to take account of dynamics and, without independent chronological data[8–10], it is difficult to interpret or reconstruct the dunefield history.

Rather than a single growth mechanism, it was recently shown[3,11] that dunes can either extend in the direction of the resultant sand flux in zones of low-sediment supply[12] (that is, the fingering mode) or grow perpendicularly to the maximum gross bedform-normal transport[13] in conditions of high-sediment supply (that is, the bed instability mode). These two distinct dune growth mechanisms determine both dune shape and orientation. Taking into account the feedback of dune aspect-ratio on the magnitude of the flow[14], these orientations can be analytically derived from the wind regime to be compared with observations in the field. Most importantly for our present purpose, the coexistence of the two dune growth mechanisms can theoretically lead to the development of different types of steady-state bedforms through spatial changes in sand availability or the emergence of secondary structures[15]. Given the apparent complexity of most dunefields on Earth, none of these interacting steady-state dune patterns have been identified and quantified so far in modern sand seas. It is an important issue because the coexistence of different bedform alignments has also been associated with changes in wind regime[16].

Raked linear dunes have been recognized for the first time in the Kumtagh desert[17] (Xinjiang province, China), where they cover an area of more than $2.4 \times 10^4\,km^2$. This remote and arid region of China has been the subject of intensive investigation in recent years[18–22]. These studies provide unique sets of data, but none of them have identified the mechanism for dune formation and the subsequent dune morphodynamics. Instead, most of them have concentrated on zibars and other grain-size segregation patterns[23], which are obvious in interdune areas.

Considering the traditional classification, raked linear dunes seem to be a specific case of compound/complex linear dunes. Their most distinguishable feature is a lateral asymmetry in the distribution of secondary ridges. At the dunefield scale, primary ridges display all the characteristics of linear dunes. They are relatively straight, extending over distances of kilometres parallel to one another with a low ratio of dune to interdune areas[24]. On the top of them, secondary ridges almost perpendicular to the main dune alignment develop only on one side of the dune. These secondary ridges have approximately the same height as the primary ridges and they resemble half crescent, so that the entire pattern is similar to a train of barchans connected to each other along the same arm.

Here we study the morphodynamics of raked linear dunes in the Kumtagh desert from two satellite images taken at a time interval of 8 years. Dune shapes, orientations and dynamics (elongation and migration) are compared with theoretical predictions derived from local wind data to show that raked linear dunes are a steady-state dune type in dynamic equilibrium with the modern wind regime. Then, we numerically and analytically investigate the conditions for the emergence of this dune type to demonstrate that the raked pattern result from the coexistence of two dune growth mechanisms.

## Results

**Morphodynamics of raked linear dunes.** Figure 1a,b shows aerial views of raked linear dunes in the Kumtagh desert (see also Supplementary Figs 1–3). At different sites in interdune areas of the Kumtagh desert, two wind towers of 2 m height have recorded the mean wind speed and direction every 15 min during 2 years from 2007 to 2009. Figure 1c shows the wind roses and the distributions of sand flux direction. This eastern end of the Tarim basin is exposed to a trimodal wind regime with a widely spread dominant wind direction from the north and two secondary peaks associated with easterly and westerly winds. Dune orientations predicted from these wind data are a north–south alignment for bedforms growing in the bed instability mode and an extension in the southwesterly direction for dunes growing in the fingering mode (Fig. 1d and Table 1 in 'Methods' section). These orientations are in general agreement with the alignments of the primary and secondary ridges observed in satellite images (Fig. 1e). These findings suggest that raked linear dunes can form by elongation and that secondary ridges can result from the development of superimposed bedforms growing perpendicularly to the maximum gross bedform-normal transport. In the framework of the two dune growth mechanisms, the two dune orientations form an oblique angle $\Delta\alpha = 70 \pm 3°$, which leads to an angle $\Delta\alpha_Q = 5 \pm 2°$ between the resultant sand flux at their crests because of the speed-up effect in multidirectional wind regimes[15] (see 'Methods' section). Then, as the superimposed bedforms nucleate and grow on the primary ridges, they migrate from one side of the linear dunes to the other. According to this migration, the regular pattern of secondary ridges is observed on the leeward slope of the primary ridges (that is, the north–western side of the linear dunes in Fig. 1). Hence, the complex asymmetric pattern of raked linear dunes may naturally arise from the coexistence of two dune growth mechanisms with both oblique orientations and dynamics.

Independently of the winds, such a scenario can be quantitatively investigated from dune morphodynamics using field data and satellite images from the Kumtagh desert (Fig. 2a,b). Figure 2c shows that the wavelength of the semi-crescentic secondary bedforms is linearly related to the height of the linear dunes. This relationship is similar to that observed for longitudinal dunes breaking in barchans under unidirectional wind regime[25–27], suggesting that the same mechanism is responsible for the singular asymmetry of raked linear dunes. In addition, the widths $W_B$ of the semi-crescentic structures exhibit a normal distribution with a mean value around 37 m and a s.d. close to 3 m across the entire dunefield (Fig. 2d). The linear dunes reach a homogeneous width $W_F$ with a mean value of 26 m (Fig. 2e) away from the growing tips, which may change shape over time (see 'Methods' section). Using a pair of Google Earth images acquired in 2008 and 2014, we estimate the elongation rate $e = 13.2 \pm 3$ m per year at the tip of linear dunes and the migration rate $c = 6 \pm 1$ m per year of the secondary ridges along the same direction (Fig. 2f). Despite more variation in the elongation rate distribution, which can be explained by the loss of sediment at the dune tips during periods of constant wind

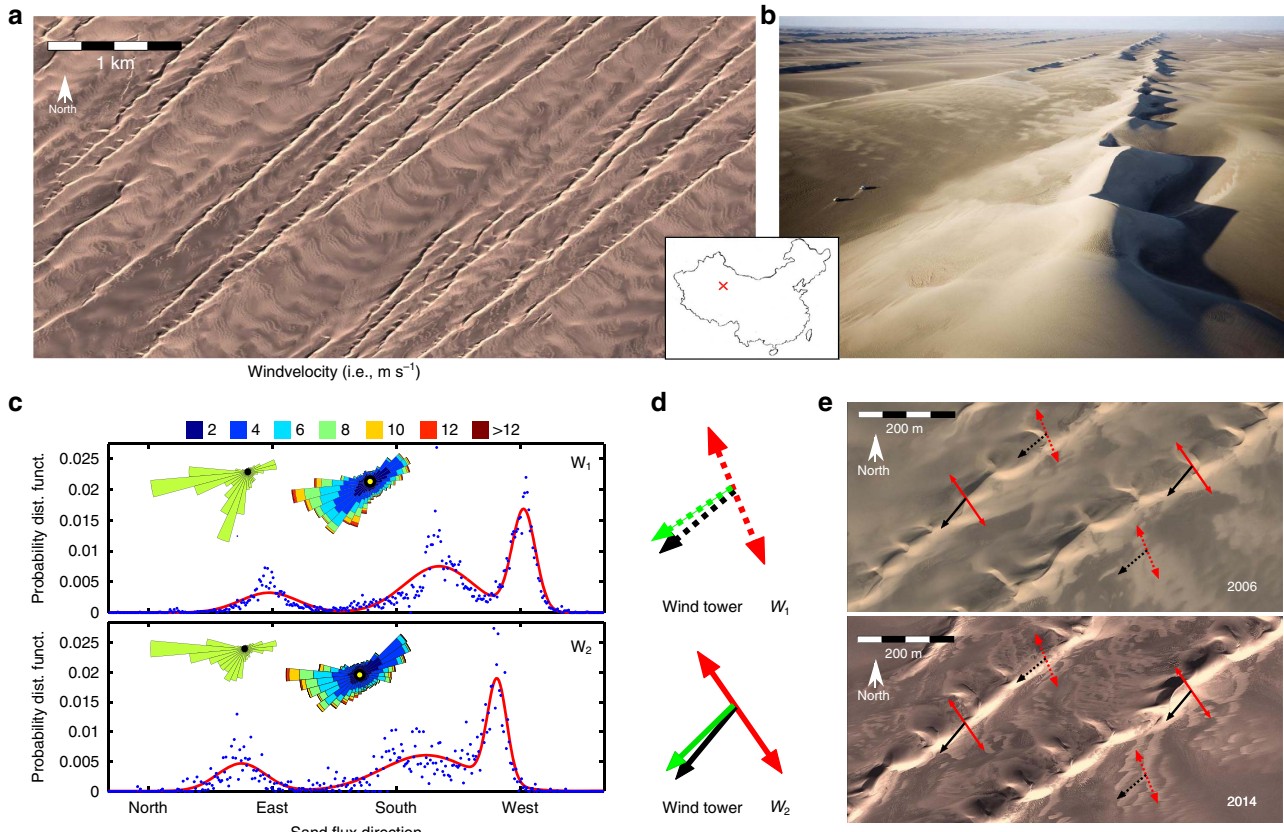

**Figure 1 | Raked linear dunes in the Kumtagh desert.** (**a,b**) Satellite and aerial views of raked linear dunes (40°13″ N, 92°13″ E). (**c**) Distributions of sand flux direction derived from the two wind towers $W_1$ (92°34′37.68″ E 40°18′32.21″ N) and $W_2$ (92°33′3.86″ E 40°15′16.43″ N) located 6 km apart in interdune areas from 2007 to 2009 (blue dots). Red curves are the best fits using a three-component Gaussian mixture model (see 'Methods' section). (**d**) Predicted orientations for dunes growing by extension (fingering mode, black arrows) and perpendicularly to the maximum gross bedform-normal transport (bed instability mode, red arrows). Green arrows show the predicted resultant sand flux at the crest of dunes growing in the bed instability mode. By definition, this is also the propagation direction of these dunes. (**e**) Comparison between predicted and observed dune orientation at two different times.

**Table 1 | Sand fluxes and dune properties derived from wind data in the Kumtagh desert.**

| Variable | Units | Wind tower $W_1$ | Wind tower $W_2$ |
|---|---|---|---|
| DP | m$^2$ per year | 47.08 | 49.02 |
| RDP | m$^2$ per year | 25.08 | 23.25 |
| $\left\|\vec{Q_I}\right\|$ | m$^2$ per year | 57.95 | 53.98 |
| $\left\|\vec{Q_F}\right\|$ | m$^2$ per year | 42.80 | 39.53 |
| RDD | ° | 230.8 | 220.1 |
| $\alpha_I$ | ° | 122.1 | 108.7 |
| $\alpha_F$ | ° | 230.0 | 220.9 |
| $\Delta\alpha$ | ° | 72.1 | 67.8 |
| $\Delta\alpha_Q$ | ° | 3.9 | 6.5 |
| RDP/DP | $\emptyset$ | 0.53 | 0.47 |
| $\sigma_F/\sigma_I$ | $\emptyset$ | 0.55 | 0.53 |

Figure 1c shows the wind and sand flux roses for the wind towers $W_1$ and $W_2$. See 'Methods' section and equations (2–10) for the description of all the variables. All angles are measured counterclockwise from East. Following Gao et al.[11], the wind speed-up have been computed with $\gamma = 1.6$. Dune orientations show little variation (<1°) with respect to the $\gamma$-value.

orientation, Fig. 2g shows that linear dunes elongate more than two times faster than the secondary ridges migrate.

**Sand fluxes on raked linear dunes.** Using the morphometric parameters of raked linear dunes, the resultant sand fluxes $Q_F$ and

$Q_B$ at the crests of primary and secondary ridges can be directly derived from the equation of conservation of mass over the entire cycle of wind reorientation. Considering that the dunes elongate or propagate without changing shape, the relations are $e = Q_F/H_F$ and $c/\cos(\Delta\alpha_Q) = Q_B/H_B$, where $\cos(\Delta\alpha_Q)$ is a correction to account for the migration direction of the superimposed bedforms in the bed instability mode and where $H_F$ and $H_B$ are the heights at the crests of primary and secondary bedforms, respectively. These heights $H_F$ and $H_B$ are derived from the measurements of the width $W_F$ of linear dunes (Fig. 2b,f) and from the width $W_B$ of half-crescentic structures (Fig. 2b,e) using the relations $H_F = (\tan(\theta_F) \times W_F)/2$ and $H_B = \tan(\theta_B) \times W_B$. Substituting in the flux equations above gives $Q_F = (e \times \tan(\theta_F) \times W_F)/2$ and $Q_I = (c \times \tan(\Theta_b) \times W_B)/\cos(\Delta\alpha_Q)$. Here we take $\theta_F = 15°$ and $\theta_B = 11°$, which are the usual reported values for the transverse aspect-ratio of linear[12] and barchan[28] dunes. Figure 3 shows the comparison between the sand fluxes computed from wind data and dune morphodynamics. The close agreement between these two independent estimates indicate that a hierarchy of bedforms can be used to provide independent quantitative constraints on sediment transport.

**Conditions for steady-state raked linear dunes.** Together, sand flux estimates (intensity and direction) and comparisons between predicted and observed dune orientations suggest that the raked linear dunes in the Kumtagh desert have reached a steady state,

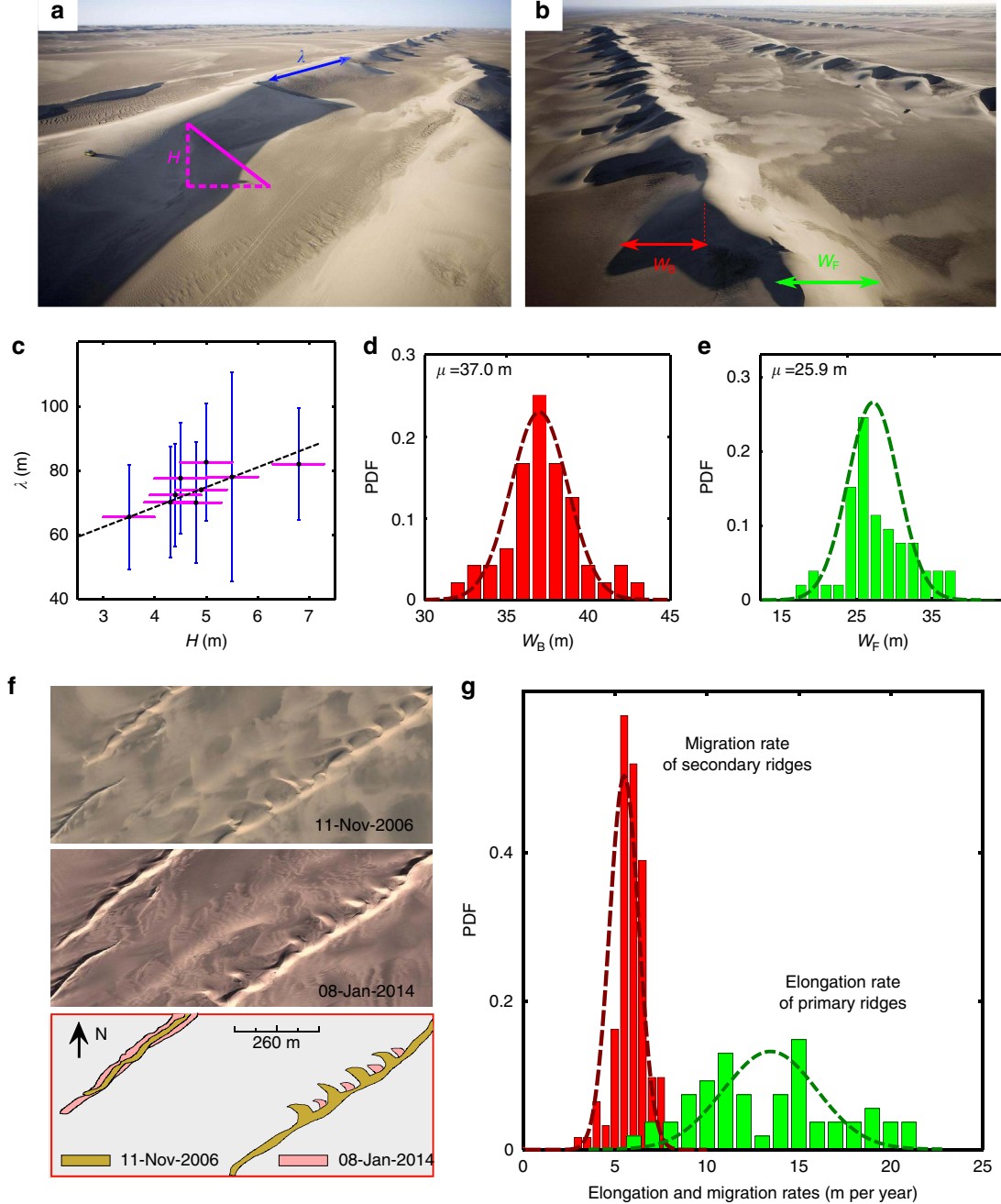

**Figure 2 | Morphodynamics of raked linear dunes.** (**a,b**) Characteristic length scales in raked linear dunefields. (**c**) Relationship between the mean wavelength $\lambda$ and the mean height $H$ of secondary ridges[17]. The line is the best linear fit with a y-intercept of $48 \pm 5$ m, a value larger but on the order of the most unstable wavelength for the formation of dunes. Errorbars are one s.d.'s in wavelength (blue) and height (magenta) measured from 50 secondary ridges. (**d,e**) Distributions of the widths $W_F$ and $W_B$ of primary and secondary ridges, respectively. (**f**) Evolution of dune shape from 2006 to 2014. (**g**) Distribution of elongation and migration rates of primary and secondary ridges over the same time period. Dashed lines in **d,e,g** are the best fits to the data using normal distributions.

which is fully consistent with the recorded wind regime. Nevertheless, such a regular pattern has never been reported before in numerical and laboratory experiments investigating dune shape[11,25,29–32]. The main reason is that all these studies have only concentrated on unidirectional and bidirectional wind regimes that seem to never satisfy the conditions for the development and stability of raked linear dunes (see 'Methods' section). From our field-based results and systematic exploration of dune shape in bidirectional wind regimes[11], we infer that raked linear dunes can only be observed in zones of low sand

availability if three conditions are met (see 'Methods' section and equations (2–10) for the derivation of all variables from wind data). First, the ratio $\sigma_F/\sigma_I$ between the dune-height growth rates of the fingering and the bed instability modes has to be large enough to promote elongation. Second, the angle $\Delta\alpha$ between the two dune orientations should be oblique, wide enough for the two modes to be distinguished and yet narrow enough to avoid segmentation of the primary ridges (especially for decreasing $\sigma_F/\sigma_I$-value). Finally, the angle $\Delta\alpha_Q$ between the migration direction of the superimposed bedforms and the elongation

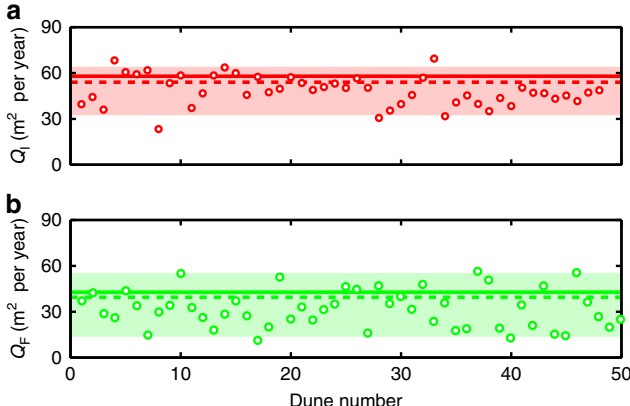

**Figure 3 | Predicted sand fluxes using two independent dune growth mechanisms.** (**a**) Resultant sand flux $Q_I$ at the crest of dunes growing in the bed instability mode. (**b**) Resultant sand flux $Q_F$ along the crest of elongating dunes. Solid (wind tower $W_1$) and dashed (wind tower $W_2$) lines are the sand flux predicted from the wind data shown in Fig. 1c. Dots are the sand flux derived from the morphodynamics of individual dunes between 2006 and 2014. The coloured areas show the 90% confidence intervals centred on the mean sand flux values $\langle Q_I \rangle = 48.3\,\mathrm{m^2 s^{-1}}$ and $\langle Q_F \rangle = 34.0\,\mathrm{m^2 s^{-1}}$.

direction should be wide enough to preserve the continuity of primary ridges and yet narrow enough for the secondary ridges and the asymmetric pattern to fully develop.

To test these hypotheses, we use a cellular automaton dune model that accounts for feedback mechanisms between the flow and the evolving bed topography[11,12,15,33–35]. Using a tridirectional wind regime similar to that observed in the field, simulations are run for $\sigma_F/\sigma_I = 0.75$, $\Delta\alpha = 42°$ and $\Delta\alpha_Q = 9.3°$. After a few cycles of wind reorientation a linear dune rapidly extends on the non-erodible bed away from the fixed source of sediment (Fig. 4). As it continues to elongate, superimposed bedforms develop and grow preferentially on the leeward side of the linear dunes according to their direction of propagation. Then, a systematic asymmetry starts to develop despite the constant loss of sediment at the tips of the linear dune and semi-crescentic structures. Over longer times, the raked pattern extends across the entire domain, keeping a constant shape characterized by regularly spaced secondary ridges of constant height and width propagating only on one side of a well-established linear dune. The tip region left appart, the superimposed dunes do not escape the finger dunes, but migrate along it. It is worth noting that the raked linear dunefield in the Kumtagh desert does not exhibit isolated barchan either. As in the numerical simulations, this suggest that the mass balance between the primary and the secondary ridges has reached an equilibrium.

## Discussion

Numerous observations of differing alignments between main and superimposed bedforms have previously been observed in both modern aeolian and subaqueous bedforms as well as in their ancient deposits. For example, Rubin and Hunter[36] and Rubin[37,38] present examples of deposit records where the two sets of bedforms coexist under (inferred) steady-state conditions. These previous studies attributed differences in orientations not to changes in flow direction through time but rather to differences between the flow generating the main bedforms (external to the dune field) and the flow producing the superimposed dunes (flow within the internal boundary layer). Here we find that, in multidirectional wind regimes, there is no

need for secondary flow or deflection of lee-side flow to give rise to the coexistence of bedforms with similar heights and different trends. There is no such ingredient in our numerical model. Instead, considering specific boundary conditions and sediment properties[23], all the independent variables required to produce the different types of dunes are incorporated into the function of flow directionality and intensity. This approach is especially justified when it comes to study primary and secondary ridges with the same height.

Our results show that raked linear dunes can be considered as a new class of composite bedforms, just as star dunes[15]. According to the usual classifications, raked linear dunes can be considered as linear dunes asymmetrically indented by a train of superimposed oblique dunes. Such a hierarchy of bedforms is possible because primary and secondary ridges are associated with different dune growth mechanisms with specific orientation and dynamics. The main dunes are in supply-limited conditions and, as a result they elongate in the direction of the resultant sand flux (fingering mode). The superimposed dunes arise from the instability of a sand bed[39] because the supply of sand from the main longitudinal dunes is essentially unlimited. Nevertheless, the simultaneous manifestation of these two modes of dune orientation occur only in limited angular ranges between crest alignments and their propagation or extension directions. As shown by numerical and analytical modelling, a tridirectional wind regime similar to that observed in the field is needed to fulfil these conditions. Unidirectional or bidirectional wind regimes cannot[11] (see 'Methods' section), which is consistent with the rareness of this dune type.

From the ratio between the elongation rate ($\approx 13$ m per year) and the length of the raked linear dunefield ($\approx 100$ km), the minimum time for the formation of the Kumtagh desert dunes under the present wind conditions is at least 5,500 years. Such a time scale and the principal wind directions are in good agreement with the presence and the alignment of yardangs at the northeastern end of the dunefield (Supplementary Fig. 4), indicating that the current wind regime may have prevailed since the Holocene thermal maximum.

More generally, this study illustrates how the combination of two dune growth mechanisms can generate new types of steady-state bedforms in dynamic equilibrium with a given multi-directional wind regime. Complex dune shapes with multiple crest orientations may develop from the coexistence of bedforms with the same[15,40] or different[3] growth mechanisms. Considering the possible number of combinations, it is impossible at this stage to generalize the dynamical behaviour observed in the Kumtagh to other environments where raked patterns have been observed. For example, there are similar dune shapes in Mali (Fig. 5a), the Namib Sand Sea[41] (Fig. 5b) and Saudi Arabia[42] (Fig. 5c). Nevertheless, these dunes are one order of magnitude larger than those in the Kumtagh desert and they develop in the middle of sand flow paths from other dune types over long time scales (>100 years) for which there is no wind data. As a consequence, their origin and stability need to be investigated further before any conclusion can be drawn. Indeed, they may also reflect transitions in dune type across space or over time[25,26]. Another interesting systematic feature to be explored is the raked dune patterns on Titan[43], the largest moon of Saturn, where trimodal wind regimes have been proposed[44,45] to explain the eastward elongation of linear dune at a global scale.

Interacting bedforms have been a challenge for decades to address major issues on the dynamics of dunefields, their origin and evolution in presence of various environmental variables[46–49]. The methodology based on the two dune growth mechanisms applies generally to dune landscapes in which different trends coexist. Here we show how it can be used to

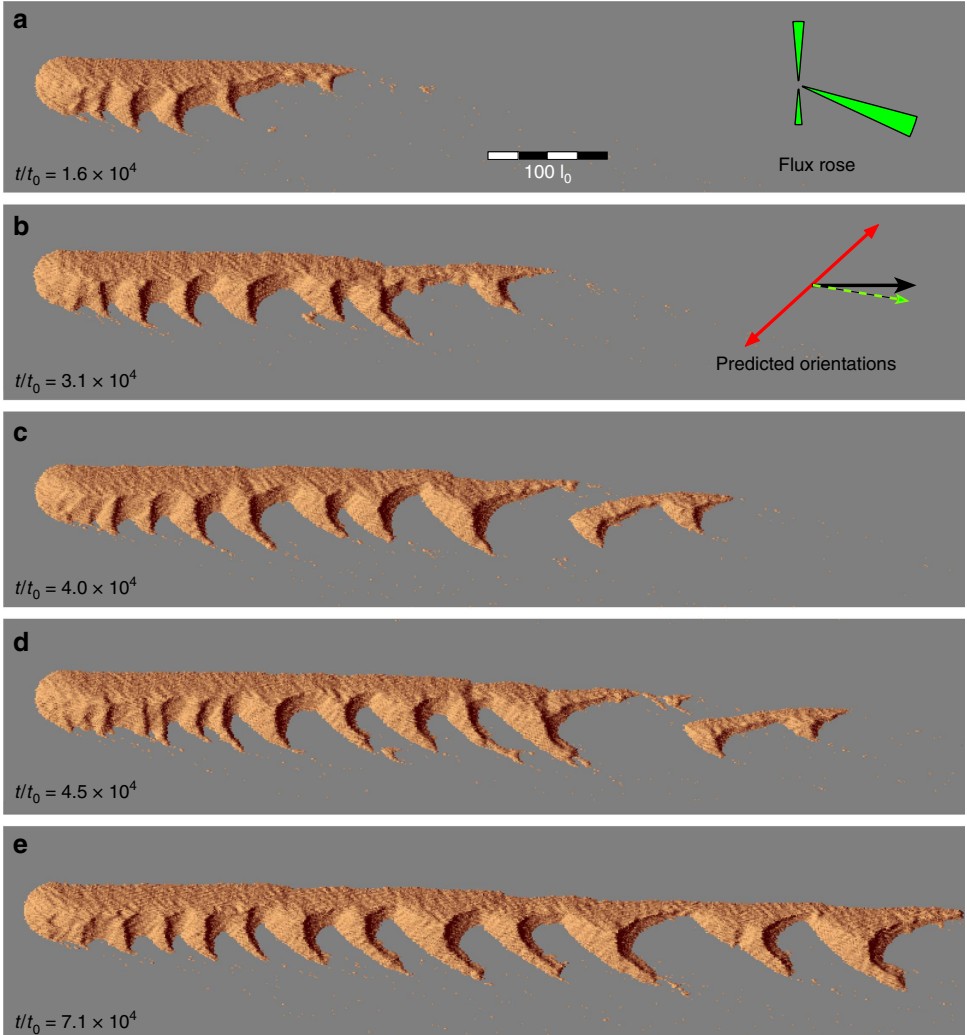

**Figure 4 | Formation of steady-state raked linear dunes from a local sand source. (a–d)** Elongating raked linear dune and development of secondary bedforms. **(e)** A snapshot of the steady-state raked linear dune. Insets show the sand flux roses and the predicted orientations of dunes growing by extension (fingering mode, black arrow) and perpendicularly to the maximum gross bedform-normal transport (bed instability mode, red arrows). The green arrow shows the resultant sand flux at the crest of dune growing in the bed instability mode. By definition, this is also the migration direction of these dunes. The asymmetric dune pattern results from the obliquity between the two dune orientations and the oblique propagation of secondary bedforms.

analyse sediment transport and local climatic conditions. Although raked linear dunes remain small-scale features ($\approx 10\,\mathrm{m}$ height) compared with giant dunes[50] ($\approx 100\,\mathrm{m}$ height), the current state of any dunefield is likely to have been reached through the coexistence of different types of bedforms, which should first be considered with respect to the two growth mechanisms. Independently, many studies have concentrated on the impact of climate change on the observed dune morphologies[16]. It is now time to combine these knowledge to deepen the understanding of modern sand seas on Earth and other planetary bodies.

## Methods

**Saturated flux on a flat sand bed.** Using the wind data collected in the Kumtagh desert (Fig. 1c), Table 1 shows the predicted sand flux on a flat sand bed and a set of variables relevant for dune morphodynamics: orientation, sand flux at the crest, migration direction and dune-height growth rate. Continuous wind measurements in the field provide the wind speed $u_i$ and direction $\vec{x}_i$ at different times $t_i$, $i \in [1; N]$. For each time step $i$, we calculate the shear velocity

$$u_*^i = \frac{u_i \kappa}{\log(z/z_0)}, \tag{1}$$

where $z = 2\,\mathrm{m}$ is the height at which the wind velocity $u_i$ has been measured, $z_0 = 10^{-3}\,\mathrm{m}$ the characteristic surface roughness and $\kappa = 0.4$ the von-Kármán constant. The threshold shear velocity value for motion inception can be determined using the formula calibrated by Iversen and Rasmussen[51]

$$u_c = 0.1 \sqrt{\frac{\rho_s}{\rho_f} g d}. \tag{2}$$

Using the gravitational acceleration $g = 9.81\,\mathrm{m\,s^{-2}}$, the grain to fluid density ratio $\rho_s/\rho_f \simeq 2.05 \times 10^3$ and the grain diameter $d = 180\,\mu\mathrm{m}$, we find $u_c = 0.19\,\mathrm{m\,s^{-1}}$, which corresponds to a threshold wind speed of $3.6\,\mathrm{m\,s^{-1}}$ two metres above the ground. For each time step $i$, the saturated sand flux $\overrightarrow{Q_i}$ on a flat sand bed can be calculated from the relationship proposed by Ungar and Haff[52] and calibrated by Durán[53]

$$Q_{\mathrm{sat}}(u_*) = \begin{cases} 25\frac{\rho_f}{\rho_s}\sqrt{\frac{d}{g}}(u_*^2 - u_c^2) & \text{for } u_* \geq u_c, \\ 0 & \text{else.} \end{cases} \tag{3}$$

Here the sand flux takes into account a dune compactness of 0.6.

From the individual saturated sand flux vectors $\overrightarrow{Q_i}$, we estimate the norm of the mean sand flux on a flat erodible bed, also called the resultant drift potential:

$$\mathrm{RDP} = \frac{\left\| \sum_{i=2}^{N} \overrightarrow{Q_i}\, \delta t_i \right\|}{\sum_{i=2}^{N} \delta t_i}, \tag{4}$$

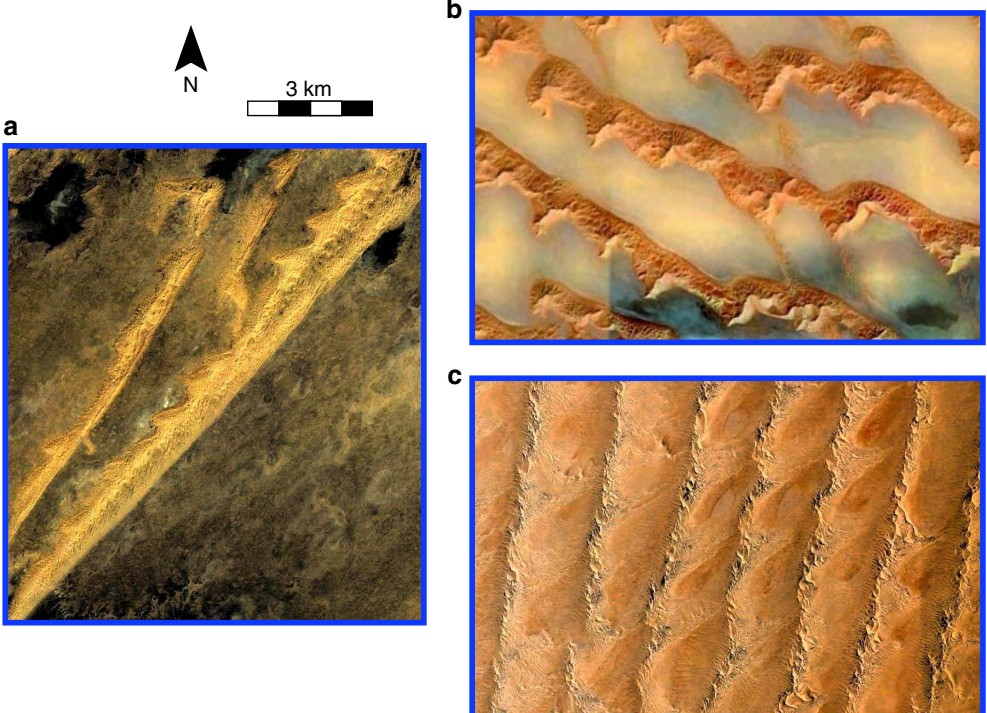

**Figure 5 | Comparison of giant linear dune features in major terrestrial sand seas.** (**a**) Erg Chech desert, Mali (24°43′ N, 3°39′ W). (**b**) Rub'al-Khali desert, Saudi Arabia (20°51′ N, 54°09′ E). (**c**) Namib desert, Namibia (24°15′ S, 14°59′ E). Despite similar raked patterns, we cannot conclude at this stage that these dunes are in a steady state or in equilibrium with the current wind regime. Given their size and location along major sand flow paths, they can also result from transition in dune shape across space and over time.

where

$$\delta t_i = t_i - t_{i-1}.$$

This quantity is strongly dependent on the function of wind directionality. We also calculate the drift potential, which is the average of individual norms and represent the mean volume of sand per unit width and time that have been transported through a vertical plane:

$$\text{DP} = \frac{\sum_{i=2}^{N} \left\| \overrightarrow{Q_i} \right\| \delta t_i}{\sum_{i=2}^{N} \delta t_i}. \tag{5}$$

Averaged over the entire time period, this quantity does not take into account the orientation of the sand fluxes[54].

The ratio RDP/DP is a non-dimensional parameter, which is often used to characterize the directional variability of the wind regimes[55,56]: RDP/DP → 1 indicates that sediment transport tends to be unidirectional; RDP/DP → 0 indicates that most of the transport components cancel each other. Finally, RDD is the resultant drift direction, that is, the direction of $\sum_{i=2}^{N} \overrightarrow{Q_i} \, \delta t_i$.

**Sand flux at the crest of dunes.** A positive topography accelerates the wind, so that the sand flux over a dune depends on the dune shape. For 2D turbulent flows over low hills, Jackson and Hunt[57] show analytically that the increase of wind velocity at the top of the hill, the so-called speed-up factor, is approximately proportional to the hump aspect ratio. Hence, at the first order of the dune aspect ratio and neglecting the transport threshold, the sand flux $\overrightarrow{Q_i^c}$ at the crest of the dune takes the following form

$$\overrightarrow{Q_i^c} = \overrightarrow{Q_i} \, (1 + \gamma |\sin(\theta_i - \alpha)|), \tag{6}$$

where $\alpha$ is the orientation angle of the linear dune, $\overrightarrow{Q_i}$ the sand flux on a flat sand bed, $\theta_i$ the orientation angle of this flux vector and $\gamma$ the flux-up ratio:

$$\gamma = \beta \frac{H}{W} \tag{7}$$

where $W$ is the width of the dune, $H$ its height and $\beta$ a dimensionless coefficient that accounts for all the other physical ingredients (for example, roughness) that affect the speed-up.

**Dune orientation $\alpha_I$ in the bed instability mode.** In the bed instability mode, dunes grow in height from the erosion of the sand bed in the interdune areas. All winds contribute to the growth of the dune height. Thus, linear bedforms develop perpendicularly to the maximum gross bedform-normal transport[13].

Considering the orientation angles $\theta_i$ of fluxes $\overrightarrow{Q_i^c}$, we calculate $Q_\perp(\alpha)$, the total sand flux perpendicular to the crest for all possible crest orientations $\alpha \in [0; \pi]$. Then, we identify the maximum value of $Q_\perp(\alpha)$ that corresponds to the most probable crest orientation $\alpha_I$ of dunes in the bed instability mode. Note that this procedure is the same as in Rubin and Hunter[13] (that is, the gross bedform-normal transport rule), except that we take into account the increase of the sand fluxes at the crest of dunes (that is, $\gamma \neq 0$). As detailed in Courrech du Pont *et al.*[3] and Gao *et al.*[11], it may significantly change the predictions of dune orientations.

**Dune orientation $\alpha_F$ in the fingering mode.** In the fingering mode, dunes extend in the direction of the mean sand flux. Hence, the orientation $\alpha_F$ of finger dunes is the one for which the direction of the mean sand flux at the crest (that is, the time average of equation (6)) aligns with dune orientation.

In practice, we calculate $Q_\perp(\alpha)$ and $Q_\parallel(\alpha)$, the total sand flux perpendicular and parallel to the crest for all possible crest orientations $\alpha \in [0; 2\pi]$. Then, we select the orientation $\alpha_F$ for which the sediment flux perpendicular to the crest vanishes (that is, $Q_\perp(\alpha) = 0$) and for which the flux parallel to the dune is positive (that is, $Q_\parallel(\alpha) > 0$). If more than one solution exists, as it is the case for star dunes[15], we look for the angle at which the $Q_\parallel$-value is maximum. By definition, when there is no feedback of topography on the flow (that is, $\gamma = 0$ in equation (6)), the orientation of the linear fingering mode $\alpha_F$ is given by the resultant sand transport direction (also called the RDD). When the wind speed-up is taken into account, the dune orientation depends on the $\gamma$-value; $\gamma = 1.6$ gives reasonable estimates of dune orientation[3,11].

From the predicted crest orientations $\alpha_I$ and $\alpha_F$ in the bed instability and the fingering modes, we calculate

$$\Delta \alpha = \alpha_I - \alpha_F, \tag{8}$$

the angle between the two bedform alignments ($\Delta \alpha \in [0; \pi/2]$).

**Sand flux at the crest of dunes in the bed instability and fingering modes.** Because the apparent dune aspect-ratio (that is, the aspect ratio seen by the wind) determines the increase in wind speed at the top of the dune, the magnitude and

the orientation of the time averaged sand flux at the dune crest $\overrightarrow{Q_{\{I,F\}}}$ is a function of the dune orientation $\alpha_{\{I,F\}}$:

$$\overrightarrow{Q_{\{I,F\}}} = \frac{\sum_{i=2}^{N} \overrightarrow{Q_i}\left(1 + \gamma\left|\sin\left(\theta_i - \alpha_{\{I,F\}}\right)\right|\right)\delta t_i}{\sum_{i=2}^{N} \delta t_i}, \qquad (9)$$

where $\theta_i$ is the orientation angle of the flux $\overrightarrow{Q_i}$ on a flat sand bed. Then, $\Delta\alpha_Q$ is the angle between the direction of the resultant sand flux at the crest of dunes in the bed instability and the fingering modes ($\Delta\alpha_Q \in [0; \pi]$).

**Dune-height growth rate.** All sand fluxes perpendicular to the crest can contribute to dune growth. Considering the dune orientation $\alpha_{\{I,F\}}$, we calculate the characteristic (growth) rate $\sigma_{\{I,F\}}$ to build up a linear dune of height $H$ and width $W$ in the bed instability or the fingering mode[3,11]:

$$\sigma_{\{I,F\}} = \frac{1}{HW} \times \frac{\sum_{i=2}^{N}\left\|\overrightarrow{Q_i}\right\|\left(1 + \gamma\left|\sin\left(\theta_i - \alpha_{\{I,F\}}\right)\right|\right)\left|\sin\left(\theta_i - \alpha_{\{I,F\}}\right)\right|\delta t_i}{\sum_{i=2}^{N} \delta t_i}. \qquad (10)$$

**Characterization of the wind regime.** We use an Expectation-Maximization algorithm to fit the flux orientation distribution by a Gaussian mixture model. Thus, we replace the real flux data by a limited number $n_\Theta$ of normal distributions characterized by a mean orientation $\Theta_i$, a standard variation $s_i$ and a weight $w_i$ with $i = \{1, \ldots, n_\Theta\}$. Considering only time periods during which the wind velocity is above a critical value for sediment transport (that is, $u_* > u_c$, see equation (3)), we assume that the probability distribution function of sand flux orientation $\Theta$ may be described by a sum of normal distributions:

$$\mathcal{P}(\Theta) = \sum_{i=1}^{n_\Theta} \frac{w_i}{s_i\sqrt{2\pi}}\exp\left(-\frac{(\Theta - \Theta_i)^2}{2s_i^2}\right). \qquad (11)$$

The Expectation-Maximization algorithm is a natural generalization of maximum likelihood estimation to the incomplete data case. Basically, this is an iterative scheme that includes two different steps. Starting from initial guesses for the parameters $w_i$, $s_i$ and $\Theta_i$, the (first) Expectation-step is to compute a probability distribution over possible completions. In the (second) Maximization-step, new parameters are determined using the current completions. These steps are repeated until convergence.

Figure 1d shows how the sand flux orientation can be fitted with a three component Gaussian mixture model ($n_\Theta = 3$). For the wind tower $W_1$, we find mean orientations $\Theta_{\{1,2,3\}} = \{250°, 189°, 13°\}$, s.d's $s_{\{1,2,3\}} = \{25°, 9°, 21°\}$ and weights $w_{\{1,2,3\}} = \{0.48, 0.35, 0.17\}$. For the wind tower $W_2$, we find mean orientations $\Theta_{\{1,2,3\}} = \{232°, 190°, 6°\}$, s.d.'s $s_{\{1,2,3\}} = \{32°, 7°, 17°\}$ and weights $w_{\{1,2,3\}} = \{0.50, 0.30, 0.20\}$. All angles are measured anticlockwise from east. The northern wind is the strongest one. The secondary wind comes from the east. The weakest wind comes from the west.

**Elongation rate and mean width of linear dunes.** To estimate the elongation and the mean width of linear dunes from Google Earth images, we isolate the contours of the growing tip at two different times (Supplementary Fig. 5a). From the surface area covered by individual dunes, we compute dune orientation and locate the dune tip at each time. Then, we measure dune width perpendicularly to this orientation at different positions along the linear dune. Supplementary Fig. 5b–e shows the variation of dune width with respect to the distance from the latest dune tip. We measure the elongation using the distance between the dune tips at two different times. We estimate the mean dune width as the homogeneous width reached along the dune body away from the growing tip. From these measurements, we observe that the shape of the growing tip changes. This is not surprising given the function of wind directionality. For this reason, we could have measured elongation using the positions at which dunes reach their homogeneous width instead of the dune tips. Nevertheless, this procedure gives similar results but more uncertainty. One should also note that the sand fluxes derived from these elongation rates are likely to be underestimated because of sand loss at the dune tips.

**No raked linear dunes under bidirectional wind regime.** Raked linear dunes have never been reported before in laboratory or numerical experiments investigating dune shape[25,29–32], even when the two dune growth mechanisms have been under consideration[3,11]. All these experiments have concentrated only on unidirectional or bidirectional wind regimes. There is clearly three dominant wind directions in the Kumtagh desert. Such a trimodal wind regime generates specific conditions, which are never met together in a bidirectional wind regime.

In the parameter space $\{\theta, N\}$ of bidirectional wind regimes ($\theta$ and $N$ are the angle and the transport ratio between the two winds, respectively), $\{\Delta\alpha, \sigma_F/\sigma_I, \Delta\alpha_Q\}$-values can be computed analytically[11] (Supplementary Fig. 6a–c). For each variable, we report the range of values derived from the wind data in the Kumtagh desert to the parameter space $\{\theta, N\}$ of bidirectional wind regimes. Thus, we

delineate different regions of the parameter space $\{\theta, N\}$. When we compare these regions, we observe that there is no overlap indicating that the specific conditions predicted from the recorded wind data cannot be reproduced by bidirectional wind regimes (Supplementary Fig. 6d–f). Therefore, there is a general incompatibility between the three variables and, even when they are considered two by two, regions of overlaps are small in size. Hence, bidirectional wind regimes cannot provide the $\{\Delta\alpha, \sigma_F/\sigma_I, \Delta\alpha_Q\}$-values met in the raked linear dunefield of the Kumtagh desert.

**Data availability.** The data that support the findings of this study are available from the corresponding author upon reasonable request. The numerical dune model has been built using the Real-Space Cellular Automaton Laboratory[35] (ReSCAL), a free software under the GNU general public licence. The source codes can be downloaded from http://www.ipgp.fr/rescal.

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

## Acknowledgements

We acknowledge financial support from the National Natural Science Foundation of China (n 41571008 and 1JY31JA51), the UnivEarthS LabEx programme of Sorbonne Paris Cité (ANR-10-LABX-0023 and ANR-11-IDEX-0005-02), the French National Research Agency (ANR-12-BS05-001-03/EXO-DUNES) and the French Chinese International laboratory SALADYN. Images of Figs 1a,e,2f and 5 are courtesy of Google Earth. Pictures of Figs 1b,2a,b are courtesy of George Steinmetz.

## Author contributions

P.L. and C.N. designed the research. Z.D. and P.L. carried out the field measurements. O.R. and C.N. developed the numerical code. P.L., C.N. and S.C.P. wrote the manuscript. All authors analysed the data and discussed the results.

## Additional information

**Competing financial interests:** The authors declare no competing financial interests.

**Publisher's note**: 

