## [Peer Review File · Nature Communications]

Reviewers' Comments:

Reviewer #1 (Remarks to the Author)

General Comments

This is an intriguing case study of how primary and secondary aeolian bedforms on different trends can co-exist in the same wind regime. The analysis is based on the authors' dune model and is supported by field wind data from the area of these dunes in the Kumtagh Desert of China. The study is a good combination of field observations and numerical modeling.

The analysis follows directly from the authors' prior work (referenced in this manuscript).

Generally the results have broad significance, although the authors make too much of their "raked linear dunes" as a special case. There are similar forms (Although the scale is different) in the Namib Sand Sea (Lancaster, 1989) and likely elsewhere (e.g. Saudi Arabia - (Holm, 1960). The manuscript would be improved and made much more relevant if the authors were to consider and analyze examples from different areas.

The conclusions are generally sound, but the impact of the manuscript would be much greater if it could be shown that the model applies generally to dune landscapes in which different trends co-exist.

The organization of the manuscript could be improved - first describe the morphology, then the wind regime that occurs, and end with the confirmation by the model.

Specific comments

Please define "raked linear dunes". These seem to be a specific case of compound/complex linear dunes.

Some of the steady state linear dune patterns have been previously identified - see Lancaster (1989).

Symbols and notation for the equations need to be defined

What is the morphologic evidence that the secondary bedforms arise on the stoss side of the dune and migrate over the crest???

Thomas' Desert Geomorphology book is too general a reference, as it contains multiple chapters by different authors - refer to a specific chapter

References

Holm, D.A., 1960. Desert geomorphology in the Arabian Peninsula. *Science*, 132, 1369-1379.
Lancaster, N., 1989. *The Namib Sand Sea: Dune forms, processes, and sediments*. A.A. Balkema, Rotterdam.

Reviewer #2 (Remarks to the Author)

A. This is an interesting case study that builds upon recent papers by the coauthors modelling dune morphodynamics demonstrating two types of dune growth mechanisms, 'fingering mode' and 'bed instability mode'. The authors use analysis of wind data to identify three winds and associated sand flux to model and interpret raked linear dune morphodynamics. They go on to suggest that this has more widespread application to complex dunefields with multidirectional wind regimes.

B. The idea that raked linear dunes are a compound dune form composed of primary and secondary linear dunes is novel. This paper is not as innovative as the coauthors modelling papers but might help geomorphologists to understand how the models can be applied in understanding complex dune morphodynamics in the field. Their approach has great potential and papers like this are required to translate the physics into a format that is more accessible to geomorphologists.

C. Data quality is good, the approach is timely and the data presentation is very good. It would help to have a map or coordinates for the wind towers. The green arrow in figure 4 is difficult to read.

D. Analysis of wind data and modelling appears to be robust.

E. The conclusion that different growth mechanisms can generate complex dune fields in dynamic equilibrium with a multidirectional wind regime is valid. The suggestion that the current wind regime may have prevailed since the "Holocene maximum" is not well substantiated and poorly expressed. They presumably mean the Holocene thermal maximum.

F. Some of the sentences construction needs attention, only minor corrections required. I was not convinced by the assertion that the "superimposed bedforms...migrate from one side of the linear dunes to the other."

G. The references are relevant.

H. The paper is well structured, the text is clear requiring only minor corrections.

Reviewer #3 (Remarks to the Author)

The manuscript by Ping et al. is an innovative and important study, but the presentation is difficult to follow. I recommend that it be accepted for publication after the authors address the following issues to clarify the presentation:

Definitions of "raked linear dunes" in Abstract, p. 3, and p. 4: Recommend giving a more precise definition of "raked linear dunes". The abstract and p. 3 mention several characteristics that are met by many dunes, but I think the authors place particular importance on the more definitive characteristic mentioned on p. 4: "These secondary ridges resemble half crescent so that the entire pattern is similar to a train of barchans connecting to each other along the same arm." If my understanding of the authors' intent is correct, this characteristic should be emphasized earlier in the ms. In any case, a precise definition should be presented near the beginning of the ms.

Abstract, various locations in the text, and caption to Fig. 4: "Secondary ridges are superimposed bedforms growing by lateral accretion⁴." The term "lateral accretion" is an odd term to use in this context. This term is usually applied to meandering river channels that do accrete laterally. I'm not sure why the authors used this term when there are already 2 other terms that are more appropriate to apply to migrating dunes. These alternatives are used on p. 3 (bed-instability mode of ref 3 and the gross-bedform normal transport rule of ref 4). If citing ref 4, the gross-bedform normal transport rule would be appropriate to use here.

Page 3, middle paragraph, last 2 sentences: Although the mechanism (for differing orientations of main dunes and superimposed dunes) proposed in this manuscript is new, numerous observations

of differing alignments between main bedforms and superimposed bedforms have previously been observed in both modern eolian and subaqueous bedforms as well as in their ancient deposits. Not all of these previous accounts attribute the differing directions to changes through time in flow as suggested here. For example, Rubin and Hunter (1983, figs. 3,6,8F), Rubin (1987, figs. 47A,B, 49, 50, 51A,B, 52A,B, 53), and Rubin (2012, figs. 1E,F, 5A) present examples where the two sets of bedforms co-exist under (inferred) steady-state conditions. These previous studies attributed differences in orientations not to changes in flow through time but rather to differences between the flow generating the main bedforms (external to the dune field) and the flow that producing the superimposed dunes (flow within the internal boundary layer) (Rubin, 2012, p. 183). Because superimposed bedforms with a different orientation from the main bedforms is such a common situation, Rubin (1987) included models of several dozen examples. The authors present a new mechanism (that likely is correct), but they should discuss the alternative that was previously suggested.

Page 4, "This eastern end of the Tarim basin is exposed to a trimodal wind regime with a widely spread dominant wind direction from the north and two secondary peaks associated with easterly and westerly winds." I don't see this pattern in the wind roses or the fitted curves. To my eye, it looks like the dominant mode in the rose plot is toward the west. Reviewing the supplemental materials, I wonder if the conclusion stated in the text is relying on the narrow standard deviation of the wind mode to the west (7°) to conclude that this mode is secondary to the mode of transport to the south (which has a standard deviation of $\sim 30^{\circ}$). My hunch is that if the modes were defined on the total amount of transport toward angular bins of any specified width (rather than within 1 or 2 standard deviations), the integrated transport mode toward the west would dominate.

Page 5: $QF = (e \times \tan(\theta_f) \times W_f)/4$: After spending considerable time reviewing this equation, my best interpretation is that QF is not the elongation flux at the crest (as stated in the text) but rather is the elongation flux averaged across the width of the dune. Assuming a triangular cross-section, flux at the crest would be double this amount (2 rather than 4 in the denominator), and flux at the edges of the dune would be \sim zero. I think the discussion of these 2 equations should be clarified in several ways. First, explain at beginning of the derivation (bottom of p. 5) where the reasoning will go (for example "Next we consider flux represented by the 2 sets of dunes to evaluate the hypothesis that ... "). Second, the expressions in the 2 equations that include aspect ratios and dune widths are used largely to substitute for the dune-height term in the common dune-flux equation (not for the speed-up effect used in the authors' previous work). It would be easier for the reader to follow this derivation if presented along the lines of: "We calculate total elongation flux for the dune from the observed volume of sediment deposited at the dune's tip (elongation rate times full dune width times dune elevation averaged from base to crest). Dune elevation varies from zero at the base to h at the crest, so mean elevation is height (h) divided by 2, so total flux at the dune tip is equal to $ewh/2$. Flux per unit width is total flux divided by the dune width over which the flux occurs ($eh/2$). Approximating dune height at the crest with dune slope (θ_f) times width of one side of the dune ($h = \tan(\theta_f) \times \text{width}/2$) and substituting in the equation above gives $QF = (e \times \tan(\theta_f) \times \text{width}/4)$." This is the same equation presented in the text, but as derived here, it gives the mean flux across the dune, not the flux at the crest. Alternatively, if the width term in this last equation is the width of one side of the dune rather than the full dune width, then this equation does, as stated, give the flux at the crest.

It is important to be clear about this derivation not only so the readers can follow it, but because the discussion compares fluxes calculated for the 2 sets of dunes, and the calculations should be comparable (i.e., comparing fluxes at the 2 kinds of crests or comparing mean fluxes for the 2 sets of dunes, and not comparing mean flux for the main dunes and crest flux for the superimposed dunes). The derivation should be clear enough that readers can understand this comparison.

Page 6: "we infer that raked linear dunes can only be observed ... if three conditions are met". The 3 specified conditions are all dependent variables that arise from wind conditions. It would be

preferable if the discussion could also consider the independent variables required to produce the observed dunes (presumably the directions and relative strengths of the 3 wind modes).

Page 7: The ms doesn't address the possibility that the flow on the two sides of the main linear dunes might vary in direction and/or strength, due to deflection of lee-side flow. The presence of such deflection has been well documented, and some readers may wonder (as I do) whether consideration of this effect might effect the ms conclusions.

Page 7: "they can be considered as linear longitudinal dunes asymmetrically indented by a train of superimposed linear oblique dunes". This is true if the superimposed dunes are oblique to transport, which may very well be true, but should be discussed more clearly in the text (what is dune orientation relative to the transport direction?).

The authors might want to cite Fenton et al. (2013). That work was done before the second mode of dune formation was known, so it is incomplete, but it nevertheless was one of the few other attempts to reconstruct multi-directional winds from bedform patterns.

In summary, this manuscript is a considerable advancement of the important and longstanding problem of interpreting winds from dune morphology. I recommend that it be accepted for publication after the authors clarify the issues listed above. I wouldn't have invested so much time in this review if it hadn't been such an interesting and educational experience.

ANSWERS TO REVIEWER 1 (R1)

General Comments

This is an intriguing case study of how primary and secondary aeolian bedforms on different trends can co-exist in the same wind regime. The analysis is based on the authors' dune model and is supported by field wind data from the area of these dunes in the Kumtagh Desert of China. The study is a good combination of field observations and numerical modeling. The analysis follows directly from the authors' prior work (referenced in this manuscript).

We would like to thank R1 for his comments and suggestions that definitely help us to improve the quality of the manuscript. As detailed below, we have taken into account all his recommendations.

Generally the results have broad significance, although the authors make too much of their "raked linear dunes" as a special case. There are similar forms (although the scale is different) in the Namib Sand Sea (Lancaster, 1989) and likely elsewhere (e.g. Saudi Arabia - (Holm, 1960)). The manuscript would be improved and made much more relevant if the authors were to consider and analyze examples from different areas.

We have added Figure 5 in the new manuscript to emphasize that raked linear dune patterns may appear in various terrestrial sand seas. Nevertheless, we have specified in a new paragraph of the discussion that, given the size of these dunes and the absence of wind data on the relevant time scale, it is impossible to conclude so far that they originate from the same morphogenic process as the raked linear dune in the Kumtagh desert. Indeed, giant dunes shown in the new Figure 5 may develop from other dune features along the local sand flow paths over time scales which are inaccessible to direct wind measurement. In other words, starting from a flat sand bed (as it is the case in the Kumtagh desert), there is no evidence that the raked pattern can develop in the different areas shown in Figure 5. Indeed, the raked pattern may also be the result from a transverse instability (see the reference to *Parteli et al. (2011)* and *Reffet et al. (2010)*) or a superposition of different sets of dunes in the fingering mode (*Zhang et al., 2012*). We have added the reference to *Lancaster (1989)* and *Holm (1960)*.

The conclusions are generally sound, but the impact of the manuscript would be much greater if it could be shown that the model applies generally to dune landscapes in which different trends co-exist.

In the last paragraph of the discussion, we have specified that

The methodology based on the two dune growth mechanisms applies generally to dune landscapes in which different trends coexist.

The organization of the manuscript could be improved - first describe the morphology, then the wind regime that occurs, and end with the confirmation by the model.

We have introduced better raked linear dunes at the beginning of the introduction and in the abstract (see also the answer to reviewer 3).

Specific comments

Please define "raked linear dunes". These seem to be a specific case of compound/complex linear dunes.

We have added a description of raked linear dunes at the beginning of the main text without using the usual classification (barchan/linear/star for dune shape; transverse/oblique/longitudinal for dune orientation) which is introduced in the next paragraph. In addition, just before the result section, we have specified that

" Considering the traditional classification, they seem to be a specific case of compound/complex linear dunes. "

Some of the steady state linear dune patterns have been previously identified - see Lancaster (1989).

In the sentence where we say that

" ... none of these interacting steady-state dune patterns have been identified and quantified so far in modern sand seas... ",

we refer to the interacting dune patterns generated by the two dune growth mechanisms within a given multidirectional wind regime. This is without any doubt an original contribution of this manuscript. We have added the reference to *Lancaster (1989)* at the beginning of the sentence where we discuss the *"apparent complexity of dunefields"*.

Symbols and notation for the equations need to be defined

All symbols and notations have been defined in the new version of the manuscript. See also the new Tab. 1 and the new Methods section.

What is the morphologic evidence that the secondary bedforms arise on the stoss side of the dune and migrate over the crest???

New measurements need to be done in the field to answer to this question. Basically, we need to look for wave-like behaviour of surface undulations on the stoss face of linear dunes according to the predicted migration direction of the superimposed bedforms.

Thomas' Desert Geomorphology book is too general a reference, as it contains multiple chapters by different authors - refer to a specific chapter

We now refer to chapter 19 of the third edition of *Thomas' Desert Geomorphology* book.

References

Holm, D.A., 1960. *Desert geomorphology in the Arabian Peninsula*. Science, **132**, 1369-1379.

Lancaster, N., 1989. *The Namib Sand Sea: Dune forms, processes, and sediments*. A.A. Balkema, Rotterdam.

We have cited these two papers in the new version of the manuscript.

ANSWERS TO REVIEWER 2 (R2)

A. This is an interesting case study that builds upon recent papers by the coauthors modelling dune morphodynamics demonstrating two types of dune growth mechanisms, 'fingering mode' and 'bed instability mode'. The authors use analysis of wind data to identify three winds and associated sand flux to model and interpret raked linear dune morphodynamics. They go on to suggest that this has more widespread application to complex dunefields with multidirectional wind regimes.

Checked.

B. The idea that raked linear dunes are a compound dune form composed of primary and secondary linear dunes is novel. This paper is not as innovative as the coauthors modelling papers but might help geomorphologists to understand how the models can be applied in understanding complex dune morphodynamics in the field. Their approach has great potential and papers like this are required to translate the physics into a format that is more accessible to geomorphologists.

Checked.

C. Data quality is good, the approach is timely and the data presentation is very good. It would help to have a map or coordinates for the wind towers. The green arrow in figure 4 is difficult to read.

We have specified the coordinates for the wind towers. We have chosen a more contrasting color for the green arrow in Figure 4.

D. Analysis of wind data and modelling appears to be robust.

Checked.

E. The conclusion that different growth mechanisms can generate complex dune fields in dynamic equilibrium with a multidirectional wind regime is valid. The suggestion that the current wind regime may have prevailed since the "Holocene maximum" is not well substantiated and poorly expressed. They presumably mean the Holocene thermal maximum.

We have specified that this period is the "*Holocene thermal maximum*".

F. Some of the sentences construction needs attention, only minor corrections required. I was not convinced by the assertion that the "superimposed bedforms...migrate from one side of the linear dunes to the other."

We have checked spelling and typos.

Because the flux direction (green arrow) at the crest of superimposed dunes is oblique to the orientation of the primary ridges (black arrows), these secondary bedforms have to migrate from one side of the linear dunes to the other.

G. The references are relevant.

Checked.

H. The paper is well structured, the text is clear requiring only minor corrections.

We have checked spelling and typos.

ANSWERS TO REVIEWER 3 (R3)

The manuscript by Ping et al. is an innovative and important study, but the presentation is difficult to follow. I recommend that it be accepted for publication after the authors address the following issues to clarify the presentation:

As detailed below, we have clarified the manuscript according to the remarks and recommendations of R3.

Definitions of "raked linear dunes" in Abstract, p. 3, and p. 4: Recommend giving a more precise definition of "raked linear dunes". The abstract and p. 3 mention several characteristics that are met by many dunes, but I think the authors place particular importance on the more definitive characteristic mentioned on p. 4: "*These secondary ridges resemble half crescent so that the entire pattern is similar to a train of barchans connecting to each other along the same arm.*" If my understanding of the authors' intent is correct, this characteristic should be emphasized earlier in the ms. In any case, a precise definition should be presented near the beginning of the ms.

We have introduced better raked linear dunes at the beginning of the main text and in the abstract (see also the answer to reviewer 1). We have clearly mentioned that "*These secondary ridges resemble half crescent so that the entire pattern is similar to a train of barchans connecting to each other along the same arm.*"

Abstract, various locations in the text, and caption to Fig. 4: "*Secondary ridges are superimposed bedforms growing by lateral accretion.*" The term "*lateral accretion*" is an odd term to use in this context. This term is usually applied to meandering river channels that do accrete laterally. I'm not sure why the authors used this term when there are already 2 other terms that are more appropriate to apply to migrating dunes. These alternatives are used on p. 3 (bed-instability mode of ref 3 and the gross-bedform normal transport rule of ref 4). If citing ref 4, the gross-bedform normal transport rule would be appropriate to use here.

Everywhere in the text and in the abstract, we have replaced the term "*lateral accretion*" either by referring to the bed instability mode or to the gross-bedform normal transport rule.

Page 3, middle paragraph, last 2 sentences: Although the mechanism (for differing orientations of main dunes and superimposed dunes) proposed in this manuscript is new, numerous observations of differing alignments between main bedforms and superimposed bedforms have previously been observed in both modern eolian and subaqueous bedforms as well as in their ancient deposits. Not all of these previous accounts attribute the differing directions to changes

through time in flow as suggested here. For example, Rubin and Hunter (1983, figs. 3, 6, 8F), Rubin (1987, figs. 47A,B, 49, 50, 51A,B, 52A,B, 53), and Rubin (2012, figs. 1E,F, 5A) present examples where the two sets of bedforms co-exist under (inferred) steady-state conditions. These previous studies attributed differences in orientations not to changes in flow through time but rather to differences between the flow generating the main bedforms (external to the dune field) and the flow that producing the superimposed dunes (flow within the internal boundary layer) (Rubin, 2012, p. 183). Because superimposed bedforms with a different orientation from the main bedforms is such a common situation, Rubin (1987) included models of several dozen examples. The authors present a new mechanism (that likely is correct), but they should discuss the alternative that was previously suggested.

We have introduced this alternative in a new paragraph at the beginning of the Discussion section.

Page 4, *"This eastern end of the Tarim basin is exposed to a trimodal wind regime with a widely spread dominant wind direction from the north and two secondary peaks associated with easterly and westerly winds."* I don't see this pattern in the wind roses or the fitted curves. To my eye, it looks like the dominant mode in the rose plot is toward the west. Reviewing the supplemental materials, I wonder if the conclusion stated in the text is relying on the narrow standard deviation of the wind mode to the west (7°) to conclude that this mode is secondary to the mode of transport to the south (which has a standard deviation of 30°). My hunch is that if the modes were defined on the total amount of transport toward angular bins of any specified width (rather than within 1 or 2 standard deviations), the integrated transport mode toward the west would dominate.

We think that it is a question of norm. We have chosen to define the primary wind according to the weight of the normal distributions using a Gaussian mixture model.

Page 5: $Q_F = (e \times \tan(\theta_f) \times W_F)/4$: After spending considerable time reviewing this equation, my best interpretation is that Q_F is not the elongation flux at the crest (as stated in the text) but rather is the elongation flux averaged across the width of the dune. Assuming a triangular cross-section, flux at the crest would be double this amount (2 rather than 4 in the denominator), and flux at the edges of the dune would be zero. I think the discussion of these 2 equations should be clarified in several ways. First, explain at beginning of the derivation (bottom of p. 5) where the reasoning will go (for example "Next we consider flux represented by the 2 sets of dunes to evaluate the hypothesis that ... "). Second, the expressions in the 2 equations that include aspect ratios and dune widths are used largely to substitute for the dune-height term in the common dune-flux equation (not for the speed-up effect used in the

authors' previous work). It would be easier for the reader to follow this derivation if presented along the lines of:

"We calculate total elongation flux for the dune from the observed volume of sediment deposited at the dune's tip (elongation rate times full dune width times dune elevation averaged from base to crest). Dune elevation varies from zero at the base to H_F at the crest, so mean elevation is height (H_F) divided by 2, so total flux at the dune tip is equal to $e \times W_F \times H_F/2$. Flux per unit width is total flux divided by the dune width over which the flux occurs ($e \times H_F/2$). Approximating dune height at the crest with dune slope (θ_f) times width of one side of the dune ($h = \tan(\theta_f)W/2$) and substituting in the equation above gives $Q_F = (e \times \tan(\theta_f) \times W_F)/4$."

This is the same equation presented in the text, but as derived here, it gives the mean flux across the dune, not the flux at the crest. Alternatively, if the width term in this last equation is the width of one side of the dune rather than the full dune width, then this equation does, as stated, give the flux at the crest.

It is important to be clear about this derivation not only so the readers can follow it, but because the discussion compares fluxes calculated for the 2 sets of dunes, and the calculations should be comparable (i.e., comparing fluxes at the 2 kinds of crests or comparing mean fluxes for the 2 sets of dunes, and not comparing mean flux for the main dunes and crest flux for the superimposed dunes). The derivation should be clear enough that readers can understand this comparison.

We agree that there was a problem in the comparison of the sand fluxes for the bed instability and fingering modes because in one case we used the sand flux at the crest (bed instability mode) and in the other the sand flux averaged over the entire width. We have homogenized both derivations distinguishing between these two fluxes. Then, we use only the sand flux at the crest for the comparisons between the sand flux properties derived from the wind data and the morphological measurements (Fig. 3). Thanks to this remark, we have changed the corresponding paragraph according to the proposed text. According to this new derivation, we have also changed the way we estimate the elongation flux and focus only on the growing tip. As a consequence, Figs. 2e, 2g and 3 have been modified and there is a new section 2 in the Supplementary information to explain how we measure elongation and the mean dune width from Google Earth images.

Page 6: *"we infer that raked linear dunes can only be observed ... if three conditions are met"*. The 3 specified conditions are all dependent variables that arise from wind conditions. It would

be preferable if the discussion could also consider the independent variables required to produce the observed dunes (presumably the directions and relative strengths of the 3 wind modes).

We have not yet explored the entire phase diagram for 3 winds, so it is presumptuous at this stage to discuss this point. Nevertheless, we have specified in the discussion that

” ... all the independent variables required to produce the different types of dunes are incorporated into the function of flow directionality.”

Page 7: The ms doesn't address the possibility that the flow on the two sides of the main linear dunes might vary in direction and/or strength, due to deflection of lee-side flow. The presence of such deflection has been well documented, and some readers may wonder (as I do) whether consideration of this effect might affect the ms conclusions.

In the first paragraph of the discussion, we have specified that we are able to produce complex dune shapes such as raked linear dunes without deflection of lee-side flow. Given the extent of the challenge, this is not the purpose of this work to explore the effect of secondary flows on dune morphodynamics. At this stage, we consider that it is a chance to address complex dune shapes with simple predictive model that do not rely on turbulence.

Page 7: "they can be considered as linear longitudinal dunes asymmetrically indented by a train of superimposed linear oblique dunes". This is true if the superimposed dunes are oblique to transport, which may very well be true, but should be discussed more clearly in the text (what is dune orientation relative to the transport direction?).

In Figures 1 and 4 of the manuscript, we have added

”Green arrows show the predicted resultant sand flux at the crest of dunes growing in the bed instability mode. By definition, this is also the propagation direction of these dunes.”

Thus, we insist one more time on obliquity between the two dune orientations and the oblique propagation of secondary dunes in the bed instability mode.

The authors might want to cite Fenton et al. (2013). That work was done before the second mode of dune formation was known, so it is incomplete, but it nevertheless was one of the few other attempts to reconstruct multi-directional winds from bedform patterns.

We have added the reference to *Fenton et al. (2013)*.

In summary, this manuscript is a considerable advancement of the important and longstanding problem of interpreting winds from dune morphology. I recommend that it be accepted for publication after the authors clarify the issues listed above. I wouldn't have invested so much time in this review if it hadn't been such an interesting and educational experience.

Thank you for the time you have spent on the manuscript. All these remarks help us to have a more general view on this broad field of research.

Reviewers' Comments:

Reviewer #1 (Remarks to the Author)

This is an interesting manuscript and much improved from the previous version.

I have many detailed comments and corrections – see the attached marked up manuscript.

The title is somewhat misleading - there is much more in this manuscript than estimating sediment flux from bedform migration and/or extension. The most significant aspect is the analysis of dune trends and the demonstration that different bedform trends can coexist.

The authors make a convincing case for why two different dune trends can co-exist in the same wind regime. However, they could emphasize the sand supply aspect of their concept – the main dunes are in a supply -limited mode – and as a result they elongate. The superimposed dunes arise because of the instability of a sand bed (see work by Elbelrhiti et al., 2005). They are transverse to the GBN because the supply of sand from the main dune is essentially unlimited.

Elbelrhiti, H., Claudin, P., Andreotti, B., 2005. Field evidence for surface-wave-induced instability of sand dunes. *Nature*, 437, 720 - 723.

Reviewer #3 (Remarks to the Author)

As I noted in my original review, this is an original and important study that considerably advances the field. I also noted that the original version had some sections that needed clarification and one equation that needed to be corrected.

i reviewed the original manuscript in considerable detail, and for the revision I read the authors' letter detailing how they addressed my review comments and read the corresponding sections in the revision, as well as quick reading of the full text. The revision fully satisfies the issues I raised, and I recommend that it be accepted for publication.

A few minor edits: There is an extra "of" in the last sentence of the middle paragraph on p. 3. and the first sentence in the last paragraph on page 3 should be "a dune" or "dunes" rather than "dune".

Finally, the last paragraph (preceding Table 1) is a good conclusion for the text. The authors might want to highlight this with a section heading "Conclusions".

I'm looking forward to being able to cite this important and innovative paper.

ANSWERS TO REVIEWER 1 (R1)

General Comments

This is an interesting manuscript and much improved from the previous version. I have many detailed comments and corrections see the attached marked up manuscript.

We have taken into account all the comments and corrections of the marked up manuscript.

The title is somewhat misleading - there is much more in this manuscript than estimating sediment flux from bedform migration and/or extension. The most significant aspect is the analysis of dune trends and the demonstration that different bedform trends can coexist.

We have changed the title in order to highlight the most important result of this study.

The authors make a convincing case for why two different dune trends can co-exist in the same wind regime. However, they could emphasise the sand supply aspect of their concept the main dunes are in a supply -limited mode and as a result they elongate. The superimposed dunes arise because of the instability of a sand bed (see work by *Elbelrhiti et al., 2005*). They are transverse to the GBN because the supply of sand from the main dune is essentially unlimited.

We have incorporated this remark in the Discussion section.

References

Elbelrhiti, H., Claudin, P., Andreotti, B., 2005. *Field evidence for surface-wave-induced instability of sand dunes*. *Nature*, **437**, 720 - 723.

We have cited this paper in the new version of the manuscript.

ANSWERS TO REVIEWER 3 (R3)

As I noted in my original review, this is an original and important study that considerably advances the field. I also noted that the original version had some sections that needed clarification and one equation that needed to be corrected. I reviewed the original manuscript in considerable detail, and for the revision I read the authors' letter detailing how they addressed my review comments and read the corresponding sections in the revision, as well as quick reading of the full text. The revision fully satisfies the issues I raised, and I recommend that it be accepted for publication.

Thank you for your effort in improving this manuscript.

A few minor edits: There is an extra "of" in the last sentence of the middle paragraph on p. 3. and the first sentence in the last paragraph on page 3 should be "a dune" or "dunes" rather than "dune".

Checked

Finally, the last paragraph (preceding Table 1) is a good conclusion for the text. The authors might want to highlight this with a section heading "Conclusions".

We have added the Conclusion section.

I'm looking forward to being able to cite this important and innovative paper.